New findings of Dunyu (Eugaleaspiformes, Galeaspida) from the Xiaoxi Formation in South China and their biostratigraphic significance

Li Qiang 1 2 3 4
Shan Xianren 3
Gai Zhikun 3 5
Chen Yang 2
Peng Lijian 1 4
Zheng Jiaqi 1 6
Lin Xianghong 6
Zhao Wenjin 3 5
Zhu Min zhumin@ivpp.ac.cn 3 5
1 Research Center of Natural History and Culture, Qujing Normal University , Qujing , China
2 Chongqing Institute of Geology and Mineral Resources , Chongqing , China
3 Key Laboratory of Vertebrate Evolution and Human Origins, Institute of Vertebrate Paleontology and Paleoanthropology, Chinese Academy of Sciences , Beijing , China
4 Key Laboratory of Yunnan Provincial Department of Education, Faculty of Biological Resource and Food Engineering, Qujing Normal University , Qujing , China
5 University of Chinese Academy of Sciences , Beijing , China
6 Yunnan Key Laboratory for Palaeobiology, Institute of Palaeontology, Yunnan University , Kunming , China
Ereskovsky Alexander
Electronic publication date: 2024 Dec 24
Publication date: 2024
Volume: 12
Electronic Location ID: e18760
Received 2024 Jul 23; Accepted 2024 Dec 4
Copyright: ©2024 Li et al.
Copyright year: 2024
Copyright holder: Li et al.
License: This is an open access article distributed under the terms of the Creative Commons Attribution License, which permits unrestricted use, distribution, reproduction and adaptation in any medium and for any purpose provided that it is properly attributed. For attribution, the original author(s), title, publication source (PeerJ) and either DOI or URL of the article must be cited.
License URL: https://creativecommons.org/licenses/by/4.0/

Keywords: Galeaspida, Dunyu, Xiaoxi formation, Biostratigraphy, Yangtze platform, South China

Funding: National Natural Science Foundation of China 42362001, 42130209 Meemann Chang Academician Workstation in Yunnan Province 202225AF150002 Yunnan Province Young and Middle-aged Academic and Technical Leaders Reserve Talents Program 202305AC350252 General Programs of the Provincial Department of Science and Technology 202101BA070001-076 Yunnan Fundamental Research projects 202201AU070017 This work was supported by the National Natural Science Foundation of China (42362001, 42130209), the Meemann Chang Academician Workstation in Yunnan Province (202225AF150002), the Yunnan Province Young and Middle-aged Academic and Technical Leaders Reserve Talents Program (202305AC350252), the General Programs of the Provincial Department of Science and Technology (202101BA070001-076) and the Yunnan Fundamental Research projects (202201AU070017). The funders had no role in study design, data collection and analysis, decision to publish, or preparation of the manuscript.

==============================
New discoveries of the late Silurian fossil fish Dunyu (Eugaleaspidae, Eugaleaspiformes, Galeaspida), Dunyu tianlu sp. nov. and Dunyu sp., are described from the Xiaoxi Formation in Xiushan of Chongqing and Xiushui of Jiangxi, China respectively. D. tianlu sp. nov. can be distinguished from D. longiforus and D. xiushanensis in its nearly equal preorbital and postorbital regions of the headshield. As the currently only known genus of Eugaleaspiformes during the late Silurian, Dunyu not only displays a large morphological difference with galeaspids from both the early Silurian and Early Devonian but also occupies a phylogenetic position that is far from the root of Eugaleaspiformes, which indicates that the lineages nested between Yongdongaspidae and Eugaleaspidae should have diversified before the early Ludlow, even during the Telychian. Discovery of new specimens of Dunyu provides direct evidence on the genus level for the correlation of the late Ludlow strata between the margin and interior of the Yangtze Platform, further supporting that the central part of the Yangtze Platform suffered from widespread transgression in the late Silurian.

INTRODUCTION

The Siluro-Devonian Galeaspida is an endemic clade of jawless stem-gnathostomes, occurring exclusively in South China, North China, and Tarim blocks (Janvier, 1996; Zhu & Gai, 2006; Janvier et al., 2009; Sansom, 2009a; Gai et al., 2018). Fossil evidence shows that Galeaspida underwent early evolutionary radiation in South China and Tarim blocks during the Telychian (Llandovery, Silurian), diversifying into three deeply rooted groups (Dayongaspidae, Hanyangaspidae, and Xiushuiaspidae), along with Eugaleaspiformes (e.g., Shuyuidae, Sinogaleaspidae, and Yongdongaspidae), as well as Polybranchiaspiformes (e.g., Gumuaspidae) (Gai et al., 2018; Shan et al., 2020; Chen et al., 2022; Shan et al., 2022a; Shan et al., 2022b; Shan et al., 2023; Zhang et al., 2024). As a result of the Kwangsian Orogeny, the Yangtze Platform of the South China Block was uplifted by the end of the Telychian (Rong, Johnson & Yang, 1984; Rong et al., 1990; Rong et al., 2019), leading to a sedimentary gap ranging from the middle-late Telychian to the late Ludlow (approximately 10 million years). Consequently, the fossil record of galeaspids during this interval is absent. By the late Ludlow of Silurian, several areas of the Yangtze Platform had developed into residual basins, where the fossil record of galeaspids reoccurred alongside other early vertebrate fossils, i.e., placoderms and osteichthyans of the Xiaoxiang vertebrate fauna (Zhu & Zhao, 2009; Zhu et al., 2009; Wang et al., 2010; Zhu et al., 2013; Zhao & Zhu, 2014; Zhao & Zhu, 2015; Zhu et al., 2016).

Dunyu, established based on the type species Dunyu longiforus from the Kuanti Formation in Qujing of Yunnan, China (Zhu et al., 2012), is the only known galeaspid fossil record from the Ludlow of Silurian. Therefore, the genus is of great significance in understanding the morphology and recovery of galeaspids during the late Silurian. Liu (1983) described Eugaleaspis xiushanensis from the late Ludlow Xiaoxi Formation (upper part of the Huixingshao or Xiaoxiyu Formation in Pan (1986)) in Xiushan of Chongqing, and this species used to represent the earliest occurrence of the genus Eugaleaspis (Liu, 1965; Liu, 1975). However, E. xiushanensis was later referred to as Dunyu xiushanensis based on the presence of the posteriorly extending cornual process, the headshield with its breadth/length ratio smaller than 1.1, and the median dorsal opening that is more posteriorly extended, a suit of characters that are absent in the Devonian Eugaleaspis (Zhu et al., 2012). Here we described a new species of Dunyu, Dunyu tianlu sp. nov., and Dunyu sp., from the Xiaoxi Formation in Xiushan of Chongqing and Xiushui of Jiangxi, China, respectively. The new discoveries not only enrich the diversity of galeaspids during the Ludlow but also provide additional evidence for the stratigraphic correlation between the Ludlow Red Beds (LDRBs) in South China.

Material and Methods

Material

The specimens of Dunyu in this study are permanently housed and accessible for examination in the collections of the Institute of Vertebrate Paleontology and Paleoanthropology, Chinese Academy of Sciences (IVPP). All fossil specimens were prepared mechanically using a vibro tool with a tungsten-carbide bit or a needle. They were measured with a digital vernier calliper, studied under optical zoom, and photographed with a Canon EOS 5D Mark III camera coupled with a Canon macro photolens (MP-E 65 mm 1:2.8 1–5 ×).

The specimens of Dunyu tianlu sp. nov. include a nearly complete headshield (IVPP V33246) and an incomplete headshield (IVPP V33247) that were collected from the Xiaoxi Formation in Tianlu scenic area of Xiushan County, Chongqing Municipality, China. The specimen of Dunyu sp. is an incomplete headshield (IVPP V30976) that was collected from the Xiaoxi Formation in Sidu Town of Xiushui County, Jiangxi Province, China. The Xiaoxi Formation, known as the Ludlow Red Beds, is mainly distributed in the interior of the Yangtze Platform of South China including central Guizhou, southeastern Chongqing, northwestern Hunan, southeastern Hubei, northwestern Jiangxi, and southwestern Anhui (Wang et al., 2010; Wang et al., 2011; Wang et al., 2017a; Wang et al., 2017b; Wang et al., 2018a; Wang et al., 2018b; Wang et al., 2018c). At both fossil localities of Dunyu, the Xiaoxi Formation shows disconformity in contact relationships with the underlying and overlying strata. The geological age of the Xiaoxi Formation is thought to be of late Ludlow, possibly extending to Pridoli, evidenced by nematophyte plants and micro-plant fossils (Wang et al., 2011; Wang et al., 2018a).

In the Xiushan area of Chongqing, the fish-bearing Xiaoxi Formation is dominated by grey-yellow and green-yellow sandstone, siltstone, silty mudstone, and mudstone (Li et al., 2021). The material of Dunyu tianlu sp. nov. was discovered from dark greenish-yellow silty mudstone near the top of the Xiaoxi Formation, approximately 9.8 m away from the bottom of the overlying Yuntaiguan Formation. In addition to plant debris, chitinozoans, and trace fossils, the associated fossils with Dunyu tianlu sp. nov. include the placoderm Bianchengichthys micros (Li et al., 2021). The Xiaoxi Formation disconformably overlies the Telychian Huixingshao Formation, in which the Chongqing Lagerstätte was found (Chen et al., 2022; Gai et al., 2022; Zhu et al., 2022).

In the Xiushui area of Jiangxi, the Dunyu-bearing Xiaoxi Formation mainly consists of yellow-green and grey-green fine-grained quartz sandstone interbedded with siltstone. The material of Dunyu sp. was collected from dark greenish-yellow silty mudstone near the top of the Xiaoxi Formation. This section is located 5 kilometers southwest of the Silurian Xikeng section. These two sections are located on the two wings of the same syncline, respectively. At the Xikeng section, the Xiaoxi Formation overlies the middle-late Telychian Xikeng Formation in which abundant galeaspids including Sinogaleaspis, Rumporostralis, and Xiushuiaspis are yielded (Pan & Wang, 1980; Pan & Wang, 1983; Gai et al., 2020; Shan et al., 2020). Dunyu sp. represents the first fossil fish discoveried in the late Ludlow Xiaoxi Formation in Jiangxi, China.

The electronic version of this article in Portable Document Format (PDF) will represent a published work according to the International Commission on Zoological Nomenclature (ICZN), and hence the new names contained in the electronic version are effectively published under that Code from the electronic edition alone. This published work and the nomenclatural acts it contains have been registered in ZooBank, the online registration system for the ICZN. The ZooBank LSIDs (Life Science Identifiers) can be resolved and the associated information viewed through any standard web browser by appending the LSID to the prefix http://zoobank.org/. The LSID for this publication is: urn:lsid:zoobank.org:pub: d7F540F5-C1A8-4C81-8599-A9004445FE3F. The LSID for Dunyu tianlu sp. nov. is: urn:lsid:zoobank.org:act:B5BE4DC7-0AE7-40F9-9FEB-DC2E3B5A29DB. The online version of this work is archived and available from the following digital repositories: PeerJ, PubMed Central and CLOCKSS.

Phylogenetic analysis

To determine the phylogenetic position of Dunyu tianlu sp. nov. within Galeaspida, an extended phylogenetic analysis based on the updated dataset of Sun et al. (2022) and Liu et al. (2023) was conducted. Two new taxa, Dunyu tianlu sp. nov. and D. xiushanensis, were added to the data matrix (Supplemental Information). The phylogenetic character data entry and formatting were performed in Mesquite (version 3.61) (Maddison & Maddison, 2015). An early plesiomorphic osteostracan Ateleaspis was selected as the outgroup for the phylogenetic analysis (Sansom, 2009b). All characters were treated as unordered and weighted equally. The dataset was subjected to the maximum parsimony analysis in the TNT software package (Goloboff & Catalano, 2016). The analysis was conducted using a traditional search strategy, with the following settings: 10,000 maximum trees in memory and 1,000 replications.

Results

Systematic paleontology

Subclass Galeaspida Tarlo, 1967	
Order Eugaleaspiformes (Liu, 1965) Liu, 1980	
Family Eugaleaspidae (Liu, 1965) Liu, 1980	

Differential diagnosis (emended). Eugaleaspidae differs from all known galeaspids in its slit-like median dorsal opening that extends posteriorly nearly to or beyond the posterior margin of the orbital opening. It differs from other families of Eugaleaspiformes in the absence of the inner cornual process.

Type genus. Eugaleaspis (Liu, 1965) Liu, 1980

Referred genera. Dunyu Zhu et al., 2012; Xitunaspis Sun et al., 2022

Genus Dunyu Zhu et al., 2012

Type species. Dunyu longiforus Zhu et al., 2012

Referred species. Dunyu xiushanensis (Liu, 1983), Dunyu tianlu sp. nov.

Differential diagnosis (emended). Dunyu differs from other Eugaleaspiformes by the cornual process that extends posteriorly, a median dorsal opening extending posteriorly beyond orbital openings, and the strong size variation of polygonal flat-topping tubercles.

Dunyu tianlu sp. nov.

Etymology. After the Tianlu scenic zone, the fossil site.

Holotype. A nearly complete headshield, IVPP V33246a, and its external mould, IVPP V33246b.

Referred specimens. An incomplete headshield, IVPP V33247.

Locality and horizon. Tianlu scenic area, Xiushan County, Chongqing, China; Xiaoxi Formation, Ludfordian, late Ludlow, Silurian.

Differential diagnosis. Dunyu tianlu sp. nov. can be distinguished from other species of Dunyu, D. longiforus and D. xiushanensis, by the following characters: small-sized headshield with a maximum length of 43.2 mm and maximum width of 51.8 mm; length ratio between preorbital and postorbital regions of headshield approaching 1.0; third lateral transverse canal without a dichotomous end.

Dunyu sp.

Material An incomplete headshield, IVPP V30976.

Locality and horizon. Sidu Town, Xiushui County, Jiangxi, China; Xiaoxi Formation, Ludfordian, Ludlow, Silurian.

Differential diagnosis. Dunyu sp. differs from other species of Dunyu by the longer preorbital region (length ratio between the preorbital and postorbital portions of headshield perhaps greater than 1.0) and the closely related orbital openings with a distance of 13.2 mm between them.

Remarks. Information about the sensory canal system and the cornual process is unknown because of the poorly-preserved specimen.

Description

Dunyu tianlu sp. nov.

The holotype IVPP V33246 (Figs. 1A and 1B) that preserves a nearly complete dorsal headshield and partially ventral headshield, together with IVPP V33247 that preserves a cornual process (Fig. 1D), enables a reconstruction of a whole headshield morphology of Dunyu tianlu sp. nov. (Figs. 1E, 2A and 2B). The headshield has a medium size with a maximum width of 51.8 mm, a midline length of 43.2 mm, and an estimated maximum length of 56.2 mm (Table 1). The width-to-length ratio of the headshield is approximately 0.91, nearly equal to that of Dunyu longiforus, which is about 0.92. The rostral margin of the headshield is blunt arciform in outline without a rostral process or a rostral angle. The headshield attains its maximum width at nearly the base of the cornual process (c) where the lateral margin of the headshield is nearly parallel (Fig. 1A). The cornual process (Fig. 1D) is spine-shaped with a total length from its base to tip of approximately 13.0 mm. The inner cornual process is absent.

Figure 1 Photographs of Dunyu tianlu sp. nov.

(A) A nearly complete internal mould of the headshield, holotype, IVPP V33246a. (B) A nearly complete external mould of the headshield, IVPP V33246b. (C) Box region of (A) showing close-up of dermal ring-like structure enclosing median dorsal opening. (D) A complete cornual process, IVPP V33247. (E) Interpretative drawings of (A). (F) Box region of (A) showing close-up of granular tubercles. Abbreviations: br.c, branchial chamber; c, cornual process; ifc, infraorbital canal; ldc, lateral dorsal canal; ltc, lateral transverse canal; mdc, median dorsal canal; md.o, median dorsal opening; obr.c, oralobranchial cavity; orb, orbital opening; pi, pineal opening; soc2, posterior supraorbital canal; vr, ventral rim.

Figure 2 Restoration of Dunyu tianlu sp. nov. in dorsal (A) and ventral (B) views.

The median dorsal opening (md.o) (Figs. 1A, 1B, 1C and 2A) is longitudinal slit-like in outline with a length of 22.9 mm and a width ranging from 1.5 mm at its middle to 1.9 mm at its anterior and posterior ends (Table 1). The posterior end of the median dorsal opening extends posteriorly beyond the level of the posterior margin of orbital openings (orb). In the holotype, the dermal exoskeleton encircling the median dorsal opening is thickened, forming a ring-like structure (Figs. 1A and 1C). The pineal opening (pi) is situated 1.9 mm away from the posterior end of the median dorsal opening (Figs. 1A, 1E, 1F and 2A), and it is small with a diameter of 0.5 mm.

The orbital openings (orb) are dorsally positioned, oval in shape (Figs. 1A, 1B, 1E and 2A), and relatively large with a long axis at 5.2 mm and a short axis at 3.61 mm (Table 1). Each orbital opening is also encircled by a dermal ring-like structure. The distance between the medial margins of two orbital openings is approximately 26.5 mm. The length of the preorbital region, from the center of the orbital opening to the rostral margin, is 21.4 mm, while that of the postorbital region, from the center of the orbital opening to the posterior margin of the headshield (excluding the cornual process), is approximately 21.8 mm (Table 1). The length ratio between preorbital and postorbital regions is nearly 1.0.

The sensory canal system, which can be observed in the internal mould of the holotype (Figs. 1A and 1E), consists of infraorbital canal (ifc), lateral dorsal canals (ldc), lateral transverse canals (ltc), posterior supraorbital canal (soc2), median dorsal canal (mdc), and dorsal commissure (dcm). The distributing pattern of the sensory canal system of Dunyu tianlu sp. nov. is strikingly similar to that of the type species D. longiforus. Specifically, the infraorbital canal stars from the anterolateral side of the orbital opening, extending posteriorly and joining with the lateral dorsal canal at a bend (Figs. 1A and 1E). The lateral dorsal canal continues posteriorly to the posterior margin of the headshield. Three lateral transverse canals (ltc1−3) issue laterally from the lateral dorsal canals, and among them, the posteriormost one is much longer than the anterior two canals (Figs. 1A and 1E). The posterior supraorbital canal starts at the anterior side of the orbital opening, extends posteriorly towards the middle line of the headshield, and connects smoothly to the median dorsal canal at the level of the pineal opening (Figs. 1A and 1E). The paired median dorsal canals are nearly parallel and converge with the opposite one to form a U-shaped trajectory. One dorsal commissure, which is roughly in level with the second lateral transverse canal (ltc2), is present to connect the median dorsal canals and lateral dorsal canals (Figs. 1A and 1E). One dorsal commissure, which connects the lateral dorsal canals and median dorsal canals, should be present, but it cannot be clearly observed in the holotype (Figs. 1A and 1E).

Table 1 Measurements and comparisons of Dunyu (mm).

Items	D. longiforus	D. xiushanensis	D. tianlu	D. sp.	
	IVPP V17681	V6793.1	V33246	V30976	
Maximum length of the headshield	85.0	55.0	56.2	—	
Maximum width of the headshield	78.0	58.0	51.1	49.2	
Length of the headshield in midline	66.0	37.0	43.2	—	
Long axis of orbital openings	10.0	5.0	5.2	4.2	
Distance between orbital openings	—	20.0	21.1	13.1	
Length of preorbital region in midline	28.0	16.5	21.4	29.8	
Length of postorbital region in midline	38.0	20.5	21.8	—	
Long axis of median dorsal opening	30.5	15.0	22.9	23.2	
Short axis of median dorsal opening	1.6∼2.5	2.0	1.5∼1.9	1.5∼2.2	
Diameter of pineal opening	2.0	—	0.5	0.9	

In the holotype, the dorsal dermal skeleton posterior to orbital openings was partially destroyed, which resulted in six pairs of branchial chambers (br.c) naturally exposed and lined by five successive arranged shallow grooves (Figs. 1A and 1E). The dorsal headshield curves ventrally to form a flat ventral rim (vr) which is partially preserved in the external mould of the holotype (Fig. 1B). The anterior portion of the ventral rim is relatively broad with a width of approximately 8.0 mm on each side. In the central area of IVPP V33246b (Fig. 1B), there is a subtriangular depression enclosed by the ventral rim, indicating the position of the oralobranchial cavity (obr.c). The oralobranchial cavity comprises an anterior oronasal cavity that opens ventrally by an oral fenestra and a posterior branchial cavity that opens ventrally by several branchial openings (Gai et al., 2011). It is noteworthy that the anterior margin of the oral fenestra, defined by the posterior margin of the anteriormost ventral rim, is in an acute angle (Figs. 1B and 2B), a condition similar to that of Dunyu longiforus and Falxcornus liui (Meng & Gai, 2021) but distinct from that of basal galeaspids such as Hanyangaspis guodingshanensis (P’an, Wang & Liu, 1975) and Changxingaspis gui (Wang, 1991) in which the anterior margin of the oral fenestra is gently arched.

The ornamentation of the headshield consists of closely set, irregular, and polygonal tubercles, a condition similar to that of Dunyu longiforus. The tubercles show various sizes in different regions of the headshield. Specifically, the tubercles in the central area of the headshield are large (Figs. 1A and 1F), with a length of 0.9 mm, whereas those around the lateral margin of the headshield are relatively smaller, with a length of 0.6 mm.

Figure 3 Photographs of Dunyu sp.

(A) An incomplete external mould of the headshield, holotype, IVPP V30976. (B) interpretative drawings of (A).

Dunyu sp.

The material of Dunyu sp. only includes an incomplete headshield with a preserved length of 46.3 mm and an estimated maximum width of 49.2 mm, suggesting that the whole length of the headshield is probably much greater than its width. The headshield width of Dunyu sp. approaches that of D. tianlu (51.1 mm) but is much smaller than that of D. xiushanensis (58.0 mm) and D. longiforus (78.0 mm). The median dorsal opening (Figs. 3A and 3B) is longitudinal slit-like in outline with a length of 23.2 mm and a width ranging from 1.5 mm to 2.2 mm (Table 1). The length-to-width ratio of the median dorsal opening is greater than 10.5. As in other species of Dunyu, the posterior end of the median dorsal opening extends posteriorly beyond the level of the posterior margin of orbital openings. The pineal opening (Figs. 3A and 3B) is far from the posterior end of the median dorsal opening with a distance of 7.0 mm between them. It is round in outline with a diameter of approximately 0.9 mm. The orbital openings (Figs. 3A and 3B) are dorsally positioned and relatively close to the midline of the headshield with a distance of 13.2 between them. The long axis of the orbital opening is nearly 4.2 mm (Table 1). The length of the preorbital region is 29.8 mm, approaching the length of D. longiforus (28.0 mm) but being much greater than the length of D. tianlu (21.4 mm) and D. xiushanensis (16.5 mm). The ornamentation of the headshield consists of closely set, polygonal, and large tubercles with the maximum length of a single tubercle exceeding 1.0 mm.

Comparison

Dunyu tianlu sp. nov. can be assigned to the genus Dunyu because it exhibits a suit of diagnostic characters of the genus, including the median dorsal opening extending posteriorly beyond orbital openings, the posteriorly extending cornual process, and no inner cornual process. D. tianlu sp. nov. is more similar to D. longiforus than D. xiushanensis in the width-to-length ratio of the headshield, which is approximately 0.9, and in the length-to-width ratio of the median dorsal opening, which is nearly 12.0. Regarding individual size, D. tianlu sp. nov. approaches D. xiushanensis but is much smaller than D. longiforus. However, D. tianlu sp. nov. markedly differs from both D. longiforus and D. xiushanensis in the length ratio between the preorbital and postorbital regions of the headshield. The ratio is 0.99 in D. tianlu sp. nov., whereas it is 0.74 in D. longiforus and 0.80 in D. xiushanensis, which means that the orbital openings of D. tianlu sp. nov. are more posteriorly positioned than those of the latter two.

The specimen IVPP V30976 resembles Dunyu, Xitunaspis, and Eugaleaspis by the longitudinal slit-like median dorsal opening that extends to the posterior margin of orbital openings. However, Xitunaspis and Eugaleaspis are known exclusively from the Lochkovian to the Pragian of Lower Devonian in Yunnan and Guangxi, whereas IVPP V30976 is collected from the upper Ludlow Xiaoxi Formation in Jiangxi. By comparison, specimen IVPP V30976 is suggestive of the late Ludlow D. tianlu sp. nov. in headshield size, the length-to-width ratio of the median dorsal opening, and the ornamentation of the headshield. It only differs from D. tianlu sp. nov. in its longer preorbital region and more closely related orbital openings. Therefore, we propose to assign IVPP V30976 to Dunyu. Considering the lack of data on the sensory canal system and the cornual process, the erection of a new species is suspended for the specimen.

Phylogenetic results

The maximum parsimony analysis produced five most parsimonious trees (Fig. 4) with a tree length = 216, consistency index (CI) = 0.389, and retention index (RI) = 0.781. A strict consensus tree shows that the monophyly of Eugaleaspidae consisting of Eugaleaspis, Xitunaspis, and Dunyu is supported by the loss of inner cornual processes (Fig. S1). Three species of Dunyu constitute a polytomy clade nested within Eugaleaspidae, supported by two synapomorphies including the posteriorly-projected cornual processes and the median dorsal opening extending posteriorly beyond the posterior margin of orbital openings.

Figure 4 A simplified phylogenetic tree of galeaspid projected against stratigraphy.

Eugaleaspiformes experienced two diversifications in Silurian and Devonian (Solid columns represent known time ranges, and thin lines represent ‘ghost lineages’) (data from (Sun et al., 2022). Abbreviations: Go., Gorstian; Ho., Homerian; Lu., Ludfordian; Sh., Sheinwoodian.

Discussion

Taxonomic implications

Dunyu tianlu sp. nov. and Dunyu sp. enriched the morphological and taxonomic diversity of Eugaleaspidae and deepened the understanding of the distribution and diversity of galeaspids during the late Silurian. Eugaleaspidae was established by Liu (1965) based on the type genus Eugaleaspis. Zhu & Gai (2006) incorporated Eugaleaspis, Yunnanogaleaspis (Pan & Wang, 1980), Pterogonaspis (Zhu, 1992), Tridensaspis (Liu, 1986) and Nochelaspis (Zhu, 1992) into Eugaleaspidae based on the first cladistically-based classification of the Galeaspida. The following described Dunyu was also assigned to the Eugaleaspidae among which Dunyu was thought to be more closely related to Eugaleaspis than to other genera by the absence of inner cornual process (Zhu et al., 2012). However, the incorporation of Yunnanogaleaspi s, Pterogonaspis, Tridensaspis, and Nochelaspis into Eugaleaspidae, as proposed by Zhu & Gai (2006), will cause the diagnosis of the family to be greatly modified to occupy a larger morphospace (e.g., bearing rostral process or not; inner cornual process absent or not; cornual process projecting laterally, posterolaterally, or posteriorly). Therefore, Shan et al. (2020) proposed to assign Pterogonaspis and Tridensaspis to the Tridensaspidae and remove Yunnanogaleaspis and Nochelaspis from the Eugaleaspidae to maintain the diagnostic stability of the Eugaleaspidae erected based on Eugaleaspis (Fig. 4). Among Eugaleaspiformes, the clade Eugaleaspidae was resolved as the highest branch by the synapomorphy of the loss of inner cornual processes, bearing a closer relationship to Tridensaspidae than Yunnanogaleaspis and Nochelaspis. The phylogenetic stability of Eugaleaspidae was also corroborated by the finding of the middle Lochkovian Xitunaspis which falls into the clade Eugaleaspidae and is sister to Dunyu (Sun et al., 2022). Dunyu tianlu sp. nov. described herein displays nearly equal preorbital and postorbital regions, which is unique among Eugaleaspidae, thus increasing the taxonomic and morphological diversity of the clade during the Ludlow of late Silurian.

Fossil records show that the Eugaleaspiformes diverged from the basal galeaspids as early as the Telychian (Llandovery, Silurian) during which they reached the highest taxonomic diversity with the occurrence of Shuyuidae, Sinogaleaspida, Yongdongaspidae, and Anjiaspis (Chen et al., 2022; Shan et al., 2022a; Shan et al., 2022b). After a major decline caused by the Yangtze Uplift, the diversity of Eugaleaspiformes gained a second peak during the Early Devonian (Fig. 4). As a stratigraphically intermediate member, however, Dunyu fails to fill the morphological gap of Eugaleaspiformes between the early Silurian and Early Devonian. By contrast, it exhibits a large number of specialized features, such as the absence of the inner cornual process and the posteriorly extending cornual process. Recent phylogenetic results resolve Dunyu as a sister group to Xitunaspis (Sun et al., 2022), a phylogenetic position that is far from the root of Eugaleaspiformes, indicating the lineages positioned between Yongdongaspidae and Eugaleaspidae should have diversified before the early Ludlow, even during the Telychian (Fig. 4).

Figure 5 Reconstructing of global plates distribution during Silurian showing the position of South China (A), paleogeographic map of South China during late Silurian (B), and the stratigraphical positions and correlations of the Ludlow Red Beds in Yangtze Region.

(A) is modified from Liu et al. (2023). (B) is based on the data from Rong et al. (2019). (C) is modified from Shan et al. (2022b). Abbreviations: I, northern Sichuan; II, western Sichuan; III, eastern Yunnan, IV, northern Jiangsu; V, southern Guangxi and Guangdong; VI, interior of the Yangtze Platform; AF, Africa, AM, Armorica, AN, Antarctic, AR, Arabia; AU, Australia; AV, Avalonia, BA, Baltica, Fm., Formation; GR, Greenland, IC, Indochina, IN, India; KA, Kazakhstan; LA, Laurentia; LRBs, Lower Red Beds; LDRBs, Ludlow Red Beds; MA, Malaya; MRBs, marine red beds; NC, North China; NG, New Guinea; SA, South America, SC, South China; SI, Siberia; TA, Tarim; URBs, Upper Red Beds.

Biostratigraphic significance

The Silurian shallow marine red beds are widely distributed in South China (Fig. 5A), and three sets of them, informally called the Lower Red Beds (LRBs), the Upper Red Beds (URBs), and the Ludlow Red Beds (LDRBs), have been recognized mainly in the following three horizons: the lower Telychian, the upper Telychian, and the upper Ludlow (Rong, Wang & Zhang, 2012; Rong et al., 2019). For a long time, the late Ludlow strata were considered to be distributed exclusively in the marginal area of the Yangtze Platform of South China Block (Rong et al., 2003), including western and northern Sichuan (Jin et al., 1989; Wan et al., 1991), eastern Yunnan (Ge et al., 1979; Wang, 2001), northern Jiangsu (Geng et al., 1997; Wang & Li, 2000; Wang & Li, 2001), and southern Guangxi and Guangdong (Fig. 5B). This paleogeographic pattern was caused by the “Yangtze Uplift” that resulted in the Yangtze Platform of South China Block as a whole being uplifted by the end of the Telychian (Llandovery, Silurian) (Rong, Johnson & Yang, 1984; Rong et al., 1990). In the past two decades, the late Ludlow shallow marine deposits (known as the Xiaoxi Formation) were successively discovered in the interior of the Yangtze Platform including Xiushan of Chongqing, Zhangjiajie of Hunan, Yinjiang of Guizhou, Yichang and Tongshan of Hubei, Xiushui of Jiangxi, and Susong of Anhui, indicating that the shallow seawater invaded into the central part of the Yangtze Platform during the late Ludlow after the “Yangtze Uplift” (Fig. 5B) (Wang et al., 2010; Wang et al., 2011; Wang et al., 2017a; Wang et al., 2017b; Wang et al., 2018a; Wang et al., 2018b; Wang et al., 2018c).

The age of the late Silurian rocks along the margin of the Yangtze Platform (Region I and III, Fig. 5B) can be determined by the conodonts like Ozarkodina snajdri and O. crispa, as well as the brachiopods like the Retziella fauna (Wan et al., 1991; Jin et al., 1992; Tang et al., 2010). For example, the Chejiaba Formation in northern Sichuan is mainly of late Ludlow age and could extend upward to lower Pridoli (Tang et al., 2010; Wang et al., 2017b). However, the common marine index fossils are nearly absent in the Ludlow deposits (Xiaoxi Formation) in the interior of the Yangtze Platform (Region VI, Fig. 5B) where the recognition and correlation of the Ludlow strata have been weakly supported by nematophytes, tubular trace fossils, as well as fossil fishes (Zhao & Zhu, 2010; Wang et al., 2017b).

In Qujing of East Yunnan, the Ludlow Red Beds (LDRBs) are found in the Kuanti Formation, which has yielded the Yangtze Vertebrate Assemblage represented by galeaspids Dunyu longiforus, placoderms Entelognathus primordialis and Qilinyu rostrata (Zhu et al., 2013; Zhu et al., 2016), and osteichthyans Guiyu oneiros and Megamastax amblyodus (Zhao & Zhu, 2010; Zhao & Zhu, 2014). This assemblage can be confidently assigned to the Ludfordian age (Ludlow, Silurian) because it is immediately beneath the first appearance of Ozarkodina crispa (Zhao & Zhu, 2014; Zhao & Zhu, 2015; Cai, Zhao & Zhu, 2020). Therefore, the Yangtze Vertebrate Assemblage can provide a palaeoichthyological standard for the correlations of the Ludlow Red Beds (LDRBs) between the margin and interior of the Yangtze Platform.

Previously, mainly based on Dunyu, the correlation between the Kuanti Formation in Qujing of Yunnan and the Xiaoxi Formation in Xiushan of Chongqing was suggested (Zhao & Zhu, 2014; Zhao & Zhu, 2015). However, the horizon of the Dunyu xiushanensis in Xiushan of Chongqing was in reality not clear in Liu (1983), and it was inferred to be the upper member of the ‘Huixingshao Formation’ (corresponding to the Xiaoxi Formation) by Pan (1986). Dunyu tianlu sp. nov. described herein confirmed that the strata yielding Dunyu in Xiushan of Chongqing belong to the Xiaoxi Formation, thus providing new evidence for the reliable correlation between the Xiaoxi Formation and the Kuanti Formation (Fig. 5C). The funding of Dunyu sp. expanded the distribution of Dunyu to northwestern Jiangxi and provided further evidence for the correlation between the Xiaoxi Formation and the Kuanti Formation in eastern Yunnan. Therefore, the genus Dunyu is of great biostratigraphic significance in the recognition and correlation of the upper Ludlow in South China (Fig. 5C). The paleogeographic distribution of the late Ludlow Dunyu in the interior of the Yangtze Platform (Fig. 6) also corroborates that the central part of the Yangtze Platform suffered from widespread transgression in the late Silurian.

Figure 6 Life restoration of Dunyu tianlu sp. nov.

Artwork credit: Jinjing Li.

CONCLUSIONS

The new materials of eugaleaspids from the upper Silurian of Chongqing, China, provide reliable diagnostic features for the erection of a new species, Dunyu tianlu sp. nov. Dunyu shows a large morphological difference to all other eugaleaspiform members and phylogenetically forms a sister to the Devonian Xitunaspis, which indicates that the lineages phylogenetically between Yongdongaspidae and Eugaleaspidae should have diversified before the early Ludlow, even during the Telychian. The occurrences of Dunyu from the Xiaoxi Formation in Chongqing and Jiangxi and the Kuanti Formation in Yunnan provide reliable evidence for the correlations of the Ludlow Red Beds (LDRBs) between the margin and interior of the Yangtze Platform of South China.

Supplemental Information

Supplemental Information 1 Strict consensus tree of 5 most parsimonious trees showing the phylogenetic position of Dunyu tianlu with Galeaspida

Tree length = 216, consistency index (CI) = 0.389, retention index (RI) = 0.781. Numbers on branches denote bootstrap frequencies (below node) and Bremer support values (above node), bootstrap frequencies below 50 are not shown.

Supplemental Information 2 Character description,

Supplemental Information 3 Data matrix for the phylogenetic analysis of Galeaspida

We thank Ridong Zhao and Jie Zhang for their assistance with the field work and Jinjing Li for the life restoration of galeaspids.

Additional Information and Declarations

Competing Interests

Author Contributions

Data Availability

New Species Registration

The authors declare there are no competing interests.

Qiang Li conceived and designed the experiments, performed the experiments, analyzed the data, prepared figures and/or tables, authored or reviewed drafts of the article, and approved the final draft.

Xianren Shan conceived and designed the experiments, performed the experiments, analyzed the data, prepared figures and/or tables, authored or reviewed drafts of the article, and approved the final draft.

Zhikun Gai conceived and designed the experiments, prepared figures and/or tables, authored or reviewed drafts of the article, and approved the final draft.

Yang Chen conceived and designed the experiments, authored or reviewed drafts of the article, and approved the final draft.

Lijian Peng conceived and designed the experiments, authored or reviewed drafts of the article, and approved the final draft.

Jiaqi Zheng conceived and designed the experiments, authored or reviewed drafts of the article, and approved the final draft.

Xianghong Lin conceived and designed the experiments, authored or reviewed drafts of the article, and approved the final draft.

Wenjin Zhao conceived and designed the experiments, authored or reviewed drafts of the article, and approved the final draft.

Min Zhu conceived and designed the experiments, authored or reviewed drafts of the article, and approved the final draft.

The following information was supplied regarding data availability:

The two headshields of Dunyu tianlu sp. nov. (IVPP V33246 and V33247) and one headshield of Dunyu sp. (IVPP V30976), are permanently housed and accessible for examination in the collections of the Institute of Vertebrate Paleontology and Paleoanthropology (IVPP), Chinese Academy of Sciences.

The following information was supplied regarding the registration of a newly described species:

Publication LSID:

urn:lsid:zoobank.org:pub:D7F540F5-C1A8-4C81-8599-A9004445FE3F.

Dunyu tianlu species LSID:

urn:lsid:zoobank.org:act:B5BE4DC7-0AE7-40F9-9FEB-DC2E3B5A29DB.

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
