# Peer review of "New findings of Dunyu (Eugaleaspiformes, Galeaspida) from the Xiaoxi Formation in South China and their biostratigraphic significance"

_PeerJ, doi:10.7717/peerj.18760_

## Round 0.1 · original submission · Minor Revisions

Dear Dr Zhu,

We have received the reports from our reviewers on your manuscript.
Based on the advice received, I feel that your manuscript could be accepted for publication should you be prepared to incorporate minor revisions.

When preparing your revised manuscript, you are asked to carefully consider the reviewers comments which are attached, and submit a list of responses to the comments. Your list of responses should be uploaded as a file in addition to your revised manuscript.

Best regards,
Alexander Ereskovsky

·

Basic reporting

Generally this meets the requirements of the journal. On the whole the English is good but there is the occasional example of inexact phrasing that would benefit from it being proofread by a colleague who is proficient in English. The Phylogenetic matrix is made available, however it is important that the character list also be made available: I do not have access to it but I understand from the text that it will be in the supplement. Figures are relevant and necessary: they could be improved by making them larger so the photographs are as big as possible, in particular figures 1 and 3. They should also include the entire phylogenetic tree. At present the figure captions are missing a lot of information, examples are listed in section 4.

Experimental design

All good: no comment.

Validity of the findings

My only concern re. validity is the phylogenetic tree. It is unusual to find only one most parsimonious tree in a big dataset and I don't think the authors have found all of the shortest trees, which affects their conclusions. I ran the matrix provided in the supplement under parsimony in PAUP and found 5 most parsimonious trees with a length of 216 steps, rather than the 1 most parsimonious tree of length 216 reported in the paper. I have attached these log file for the analysis showing what I did as a pdf . This doesn’t affect the tree topology very much, but it does mean that Dunyu breaks down into a polytomy in the strict consensus, which would affect conclusions about the interrelationships of Dunyu species. I recommend the authors try a more extensive search in tnt or try the matrix in paup to make sure they’ve surveyed tree space sufficiently, and amend their conclusions accordingly.

Additional comments

Overall this paper is a nice and to-the-point reporting of a new fossil species and I recommend it is published once the authors have address my major issue, which is with the phylogenetic analysis. A couple of other more important points about figures are below.
1. Photographs in figures could be made bigger to allow the reader to see anatomical details, e.g. the squamation on the thickened rim of the nasal opening.
2. The full phylogeny should be published as a supplementary figure: at present only a summarized version is presented where some of the tips are collapsed.


Line-by-line comments on the paper.
Lines 32 and 412: “shows a large morphological disparity” Disparity is morphological diversity so is a trait of a group rather than a specific taxon. Better to say that Dunyu expands the disparity observed in Eugaleaspida?
Line 42: “strongly endemic”. Can something be “strongly” endemic? Presumably it is endemic or it isn’t? Recommend dropping “strongly”
Line 47: “plesiomorphic group” Plesiomorphic describes a morphological character rather than a group. A group can have lots of plesiomorphic characters if that’s what the authors are trying to say?
Line 126: Selection of Ateleaspis as an outgroup should be justified and referenced (I guess going off Sansom 2009?
Line 139: “silt-like” should be “slit-like”
Line 198: “thickened, forming a ring-like structure”. Does this correspond to a specific morphotype of dermal scale/tessera? Would be nice to have a close-up image in the figure, as it’s a bit hard to see
Line 306: Might be interesting to outline what the character is that unites the other two species of Dunyu to the exclusion of D. tianlu (although see comments on phylogenetic analysis).

Figures
Figure 1. This figure caption requires the specimen number and abbreviations listing.
Figure 1 + 3. Both of these could use the maximum figure size (1 page) to make the photos as big as possible.
Figure 2. This figure caption requires the specimen number and abbreviations listing.
Figure 3: The photograph in this looks quite low quality, although it might be the pdf’s fault. This figure caption requires the specimen number and abbreviations listing.
Figure 4. How was the time calibration of this cladogram calculated?
Figure 5: are the red bits of the figure the “red beds”? This needs to be made explicit in the caption.

·

Basic reporting

The paper is well wrotten, with a clear English language,and using professional terms of Paleozoic fossil fish. Literature references are sufficiently cited, and field background is stated in details. The article is well organized with a professional structure, suitable figures and tables of fossil fish research. Raw data are shared as supplementary materials that can be easily checked.

Experimental design

The introduction part clearly state the research history and advances, so that meaningful research questions are well defined. Material and methods are stated in details, which make those questions resolve possible.

Validity of the findings

The article reported on two new late Silurian fossil fish Dunyu tianlu sp. nov. and Dunyu sp. from the Xiaoxi Formation in Xiushan of Chongqing and Xiushui of Jiangxi, respectively. They are described in details and thoroughly compared with other species of the genus Dunyu, among which Dunyu tianlu sp. nov. was phylogenetically analysed. New findings enrich the diversity of galeaspids during the Ludlow but also provide potential biostratigraphic significances.

Additional comments

Any other comments are cited in PDF attachment. Please check them.

·

Basic reporting

The manuscript by Li et al. provides new and important taxonomic and phylogenetic information on early galeaspids from China. These are an important group of vertebrates, not least because they occupy a phylogenetic position close to the origin of key gnathostome traits such as jaws and paired fins.

The english syntax within the manuscript could be improved. There are a number of sections, particularly in the introduction, that are phrased strangely. For example:

“Fossil records show that Galeaspida gained its early evolutionary radiation in the South China and Tarim blocks during the Telychian (Llandovery, Silurian) with the diversification of three plesiomorphic groups (Dayongaspidae, Hanyangaspidae, and Xiushuiaspidae), Eugaleaspiformes (e.g. Shuyuidae, Sinogaleaspidae, and Yongdongaspidae), as well as Polybranchiaspiformes (e.g. Gumuaspidae) (Gai et al., 2018; Shan et al., 2020; Chen et al., 2022; Shan et al., 2022a; Shan et al., 2022b; Shan et al., 2023; Zhang et al., 2023).”

"“Dunyu, which is erected based on the type species Dunyu longiforus from the Kuanti Formation in Qujing of Yunnan, China (Zhu et al., 2012), is the only known galeaspids during the Ludlow of Silurian."

The manuscript is well cited with relevant literature. There was a notable omission in the opening sentence:

"The Siluro-Devonian Galeaspida is a strongly endemic clade of jawless stem-gnathostomes, occurring exclusively in the South China, North China, and Tarim blocks (Janvier, 1996; Zhu and Gai, 2006; Janvier et al., 2009; Gai et al., 2018)".

- You should cite Sansom 2009 “Endemicity and palaeobiogeography of the Osteostraci and Galeaspida: a test of scenarios of gnathostome evolution”.

The manuscript is professionally structured with excellent figures and represents a self-contained body of work.

Experimental design

The manuscript falls within the Aims and Scope of the journal.

The research question is well defined. Although some clarity is needed in the introduction:

“Due to the Kwangsian Orogeny, however, the Yangtze Platform of the South China Block as a whole was uplifted by the end of the Telychian (Rong et al., 1984; 53 Rong et al., 1990; Rong et al., 2019), resulting in the disappearance of a shallow marine environment suitable for galeaspids living. Because of this, the galeaspids underwent a rapid decline in diversity and remained unknown until the late Ludlow of Silurian”

- This section seems to imply that the apparent decline in galeaspid diversity during the Llandovery-Ludlow is a genuine pattern resulting from extinctions due to habitat loss. However, an alternative and perhaps more parsimonious explanation is that the decline is an artefact of the rock record, given the absence of shallow marine facies during this interval. I think some further clarification is required here.

The research presented here is completed to a rigorous technical standard and the methods are sufficiently detailed for the phylogenetic analyses or taxonomic work to be replicated.

Validity of the findings

All the underlying data are provided. The conclusions drawn are supported by the results of the taxonomic and phylogenetic analyses presented in the manuscript.

Additional comments

I look forward to seeing this work published.

---

## Round 0.2 · accepted · Accept

Dear Dr. Zhu,

You have substantially improved your manuscript following the reviewers' comments and responded to all of their comments. Your manuscript is now ready for publication.

All the best,
Alexander Ereskovsky

·

Basic reporting

The authors have done a good job addressing issues with English language in the manuscript and have enlarged figure images.

Experimental design

As before the experimental design is good.

Validity of the findings

The authors have addressed my concerns with the phylogenetic analysis.

Additional comments

I am satisfied that the authors have addressed my comments and, as far as I can tell, those of other reviewers. I think this is a nice contribution to the literature which I look forward to seeing published. Congratulations to the authors!

·

Basic reporting

The article is well arranged in professional English. It structure, figures and tables are professional, and conforming to the principle of systematical palaeontology. The references cited in the text are enough to support the results.

Experimental design

This new fossil findings and its taxonomy well fit the aims and scope of the comprehensive Peerj. The research questions are meaningful, with broad interests.

Validity of the findings

The impact and novelty of findings in this article are well assessed, and research results are positive with enough fossil evidence.

Additional comments

Authors have answered all comments that I mentioned in the my first review. I appreciated the authors' efforts and satisfied with this well arranged article. The findings and its scientific meanings are of very important so that it well fit the aim and scope of Peer J.